# SPLIT DECISIONS: VLM-GUIDED ACTION SAMPLING FOR EFFICIENT RL EXPLORATION

## ABSTRACT

Reinforcement learning (RL) offers a general framework for adapting vision-language-action models (VLAs) to new tasks, but its effectiveness is often bottle-necked by inefficient exploration. Existing strategies waste interactions on uninformative behaviors, hindering sample efficiency. We introduce Split Decisions, an exploration framework that leverages semantic priors from vision-language models (VLMs) to guide VLAs toward more promising regions of the action space. In our approach, the VLM serves as a high-level planner that proposes subgoals, while the VLA acts as a low-level controller that samples and executes actions aligned with those subgoals. This structured guidance improves both the efficiency and quality of exploration, enabling policies to discover rewarding strategies more quickly. We evaluate Split Decisions on robotic manipulation tasks in SimplerEnv under both online and offline RL settings. In online fine-tuning, it achieves up to a 31% gain in task success with the same interaction budget, while in offline training, datasets collected with Split Decisions provide a 27.5% improvement over prior methods. These results establish Split Decisions as a general and effective paradigm for enhancing exploration in VLA adaptation.

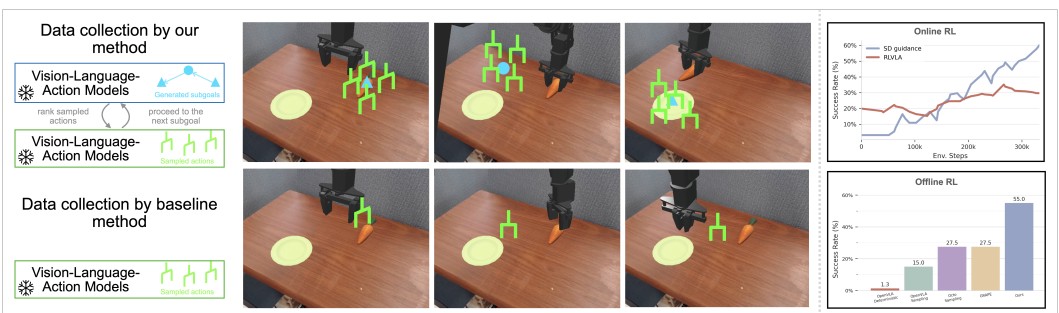

Figure 1: **Overview of Split Decisions.** Our approach leverages semantic priors from vision-language models (VLMs) to guide vision-language-action models (VLAs) toward more promising regions of the action space. A VLM generates subgoals that structure exploration, while the VLA samples and executes aligned actions. This yields higher-quality demonstrations without human supervision and enables more efficient learning. Experiments show improved success rates in both online and offline RL compared to baseline exploration methods.

## 1 INTRODUCTION

Reinforcement learning (RL) enables automatic discovery of optimal strategies via trial-and-error in the environment, providing a general framework of training policies from scratch (Zhu et al., 2019; Yu et al., 2020) or fine-tuning pre-trained policies (Ball et al., 2023) for decision making. Recent works have demonstrated online RL facilitates effective adaptation of large language models (Rafailov et al., 2023a), vision-language models (VLMs) (Zhai et al., 2024) and vision-language-action models (VLAs) (Guo et al., 2025) to novel tasks or environments. Yet, RL is notoriously sample inefficient. To discover successful strategies, it often requires millions, if not billions, of samples (Mnih et al.,

2013). This fundamentally limits RL's applicability to tackling long-horizon, sparse-reward, and complex-dynamic tasks, such as robotic manipulation (Levine et al., 2018).

To collect data efficiently and effectively, also known as the *exploration* problem, stands at the core of reinforcement learning. A wide range of exploration methods have been proposed, such as random exploration (Sutton et al., 1998), curiosity-driven exploration (Pathak et al., 2017), knowledge-based exploration (Houthooft et al., 2016). However, these methods do not leverage any prior knowledge, but merely rely on inductive biases, and consequently fail to address the core challenge of exploration. In practice, policies are still tasked to select arbitrary, low-value actions, which are often uninformative, before discovering high-value behaviors.

How can we explore more intelligently? Our key insight is to leverage semantic prior knowledge to guide exploration. We draw inspiration from human behavior: when faced with a novel task, humans do not rely on random trial-and-error. Instead, they intuitively select a set of plausible actions informed by prior experience or common-sense knowledge. Meanwhile, VLMs have shown strong capabilities in encoding semantic and commonsense knowledge (Achiam et al., 2023; Team et al., 2025). We are motivated to use VLMs to guide the exploration of an RL robot policy.

Modulating a robot's action sampling with VLMs presents three fundamental challenges: (1) VLMs lack precise 3D grounding, making it difficult to associate predicted 3D actions with the scene geometry; (2) there is no principled interface between VLMs and robot policies: while VLMs output natural language, robotic control requires high-precision continuous values, and naïvely mapping between the two is unreliable; (3) recent attempts to use VLMs for reward shaping in RL (Patel et al., 2025) fail to address the exploration problem. Like typical exploration methods, they still rely on repeatedly sampling low-value actions before stumbling upon high-value ones.

We introduce Split Decisions, which leverages VLMs to predict spatio-temporal subgoals that directly guide action sampling toward promising regions of the action space. Split Decisions contains a VLM functioning as a high-level planner to determine *how* to explore, and a low-level control policy, such as a VLA, generating actions to execute the exploration. Conditioned on a task instruction, the VLM produces a sequence of subgoals necessary for completing the task, along with completion conditions that define successful exploration for each subgoal. In this work, we focus on robotic manipulation, and represent subgoals as 2D waypoint trajectories (Gu et al., 2023; Wen et al., 2023; Ji et al., 2025). To explore a given subgoal, the low-level policy samples multiple random actions, scores them based on their spatial proximity to the subgoal, and executes a top-ranked action. Upon satisfying the current subgoal's completion condition, the system proceeds to the next. This formulation enables the VLM to reduce temporal complexity by decomposing long-horizon tasks into sequential subtasks, while simultaneously reducing spatial complexity by constraining the action search space via affordance priors. Together, these components lead to more efficient and effective exploration.

We test Split Decisions in the setup of fine-tuning VLAs with online and offline RL, while Split Decisions demonstrates superior sample efficiency. Using the same total number of interaction steps, Split Decisions achieves a 31% absolute performance gain against previous state-of-the-art (Liu et al., 2025), for online RL fine-tuning OpenVLA (Kim et al., 2024) on a simulation benchmark extended from BridgeData. Meanwhile, under the offline RL setting, Split Decisions outperforms prior art (Zhang et al., 2024) by an absolute performance gain of 27.5% on BridgeData (Li et al., 2024c) simulation benchmark. Moreover, our method goes beyond simple pick-and-place tasks, generalizing to other manipulation tasks. It improves the multi-task performance of 'Close drawer' and 'Open drawer' task by 40.8% on Google Robot simulation benchmark (Li et al., 2024c).

## 2 RELATED WORK

**Exploration in RL.** Exploration aims to discover environmental information, such as high-value actions, by trying out different actions (Kolter & Ng, 2009). The objective of conventional approaches are based on different types of inductive bias, including policy randomness (Sutton et al., 1998), maximization of curiosity (Oudeyer et al., 2007; Pathak et al., 2017) or information gain (Houthooft et al., 2016), and goodness of the initial state (Ecoffet et al., 2019). Other techniques are also proposed to enhance exploration in RL, such as curriculum learning (Wan et al., 2023; Yin et al., 2025), reward shaping (Andrychowicz et al., 2020; Kumar et al., 2021) and knowledge distillation from expert

demonstrations (Rajeswaran et al., 2017; Nair et al., 2018; Gupta et al., 2019). However, all these methods are still sample inefficient, as futile low-value actions are sampled during exploration.

To enhance sample efficiency, recent work leverages semantic priors of VLMs to reduce the action search space during exploration. VLMs are prompted to generate texts (Liu et al., 2024b; Lee et al., 2025) or affordance (Ma et al., 2024), which are then transformed into robot actions. In contrast, Split Decisions marries VLMs and VLAs, where semantic priors of VLMs bias action sampling of VLAs. Such an integrated system can generate more precise action control than previous methods.

**Vision-language-action models.** While VLAs adopt the same architecture as VLMs, they are fine-tuned to map multimodal inputs to executable actions. Such training recipes enable VLAs to predict high-precision action control for dynamic manipulation tasks. Early works such as RT-1/2 (Brohan et al., 2022; 2023) and PaLM-E (Driess et al., 2023) validate this paradigm at scale. Follow-up works have shown different types of improvement over the original VLA models, including increased task diversity (Black et al., 2024; Intelligence et al., 2025), enhanced reasoning ability (Li et al., 2024a), control precision (Qu et al., 2025), and public accessibility (Kim et al., 2024).

While VLAs have shown promising results in similar domains as training distributions, these models are incapable of generalizing to previously unseen environments. As a result, adapting these models to novel scenes by fine-tuning has attracted increasing attention. For instance, Mark et al. (2024); Chen et al. (2025); Guo et al. (2025); Liu et al. (2025); Zhang et al. (2024) adopt online RL frameworks to optimize VLAs. Our method is complementary to these approaches on adaptation, as Split Decisions aims to improve sample efficiency of online RL.

**Vision-language models in robotics.** The integration of VLMs into robotics has recently attracted growing attention. A common strategy is to adapt VLMs into robot policies. While VLMs can be prompted directly to generate actions in text form (Team et al., 2025; Zhou et al., 2025), textual outputs are inherently ill-suited for representing continuous control signals, limiting their precision in fine-grained action execution. To address this, several approaches have been explored, reformulating action generation as the problem of executable code generation (Liang et al., 2023; Lin et al., 2023), visual prompts selection (Nasiriany et al., 2024; Liu et al., 2024a), and affordance-based motion planning (Huang et al., 2023; Duan et al., 2024). Despite these advances, VLM-based policies still underperform compared to models explicitly trained for robotic manipulation, as VLMs are not pre-trained on large-scale robotic tasks. In contrast, Split Decisions does not rely on VLMs to directly generate action controls; instead, it leverages VLMs to guide the action sampling of low-level robot policies during training.

## 3  PRELIMINARIES

**Problem formulation.**  We formulate robotic manipulation as a Markov Decision Processes (MDP) defined by a tuple $(S, A, P, R, \gamma, p_0)$, where $S$ is the state space representing robot and environment configurations, $A$ is the action space, $P : S \times A \times S \to [0, 1]$ is the transition probability function, $R : S \times A \to \mathbb{R}$ is the reward function, $\gamma \in [0, 1]$ is the discount factor, and $p_0$ is the initial state distribution. Let $\pi_\theta : S \times A \to [0, 1]$ denote a policy parameterized by $\theta$ that outputs a probability distribution over actions given input states. The objective is to optimize the policy by maximizing the expected return $\mathbb{E}_{\tau \sim \rho_\theta}[\sum_{t=0}^{T-1} \gamma^t r_t]$, where $\tau = \{(s_t, a_t, r_t)\}_{t=0}^{T-1}$ represents a sequence of state-action-reward tuples over a finite horizon $T$, and $\rho_\theta$ denotes state-action marginals of the trajectory distribution induced by policy $\pi_\theta$, transition function $P$ and initial state distribution $p_0$.

**Online RL and PPO.**  Online RL alternates between exploration and policy learning (Sutton et al., 1998). In the exploration phase, actions sampled from the policy are executed in the environment, producing a trajectory $\tau$ that is stored in a replay buffer $B$. In the learning phase, the policy $\pi_\theta$ is updated using trajectories from $B$.

Proximal Policy Optimization (PPO; Schulman et al., 2017) is a general, effective framework of online RL, which adopts the following loss:

$$L_{PPO}(\theta) = -\mathbb{E}\left[\min\left(\frac{\pi_\theta(a_t|s_t)}{\pi_{old}(a_t|s_t)}\hat{A}_t, \text{clip}\left(\frac{\pi_\theta(a_t|s_t)}{\pi_{old}(a_t|s_t)}, 1-\varepsilon, 1+\varepsilon\right)\hat{A}_t\right)\right] \quad (1)$$

where $\hat{A}_t$ denotes an estimator of the advantage function at timestep t and $\varepsilon$ is a clipping threshold.

Figure 2: **Framework of Split Decisions.** Split Decisions guides RL exploration by using VLMs to generate waypoint subgoals and VLAs to propose candidate actions. Actions closest to the subgoals are selected, enabling more efficient exploration and higher-quality rollouts for policy learning.

**Offline RL and DPO.** Offline RL (Levine et al., 2020) considers an easier setup that learns a policy entirely from a fixed dataset, without any further interaction with the environment during training. In our case, the dataset is collected using Split Decisions with a pre-trained policy (OpenVLA (Kim et al., 2024)), which is then fine-tuned on the collected data.

Direct Preference Optimization (DPO) (Rafailov et al., 2023b) provides a preference-based formulation of offline RL, where rewards are derived from pairwise rankings of trajectories (Bradley & Terry, 1952). In this work, successful trajectories are preferred over failed ones, yielding the following loss:

$$L_{DPO}(\theta) = -\mathbb{E}_{(s,a^+,a^-)} \left[ \log \sigma \left( \beta \left( \log \frac{\pi_\theta(a^+|s)}{\pi_{ref}(a^+|s)} - \log \frac{\pi_\theta(a^-|s)}{\pi_{ref}(a^-|s)} \right) \right) \right] \quad (2)$$

where $\pi_{ref}$ is the reference policy used for data collection, $a^+$ and $a^-$ are actions from successful and failed trajectories, respectively, $\sigma(\cdot)$ is the sigmoid function, and $\beta$ is a scaling factor. This objective aligns the learned policy $\pi_\theta$ toward preferred behaviors relative to $\pi_{ref}$.

In our experiments, agents receive only visual observations $o_t$ at each timestep $t$, rather than the full environment state $s_t$. Accordingly, in the following sections we use $o_t$ in place of $s_t$.

## 4 METHOD

We introduce Split Decisions, a novel framework enhancing RL exploration for robotic manipulation. Specifically, it guides action sampling toward semantically meaningful regions using spatio-temporal subgoals predicted by a VLM. As a result, Split Decisions is capable of sampling higher-value actions and gathering higher-quality trajectories than typical RL exploration. While Split Decisions integrates seamlessly with both online and offline RL algorithms, we identify two challenges in RL fine-tuning a state-of-the-art policy, OpenVLA, with the guidance of our method: (1) action sampling is computationally expensive, and (2) unimodal output distribution of OpenVLA makes guided RL fine-tuning ineffective. To overcome these limitations, we propose augmenting action sampling with Gaussian noise, along with revisions in online and offline RL pipeline. Finally, Split Decisions is used only during training to collect interaction data; at inference, the policy runs independently. Algorithm 1 summarizes the entire exploration pipeline. See Fig. 2 for the overview of our framework.

### 4.1 VLM-GUIDED EXPLORATION

In the exploration phase of RL, Split Decisions performs action sampling in two stages: (1) sub-goal planning, and (2) guided action sampling. At the start of an episode, a VLM generates spatio-temporal subgoals–2D waypoint trajectories (Gu et al., 2023)–conditioned on the current visual observation and language instruction. Split Decisions then explores these subgoals sequentially: for each subgoal, it samples multiple candidate actions from the policy, ranks them by proximity to the subgoal, and executes one selected from the top-ranked set. When a subgoal is achieved, Split Decisions advances

to the next; otherwise, it continues exploring the current subgoal until the episode ends. We now describe these two stages in detail.

**Subgoal planning.** The central idea of Split Decisions is leveraging VLMs' 2D visual reasoning and task planning to guide 3D robot control. Given a task instruction, the VLM decomposes the task into subtasks, identifies relevant image regions, and proposes target end-effector actions as subgoals. Since current VLMs lack reliable 3D grounding (Zhang et al., 2025), Split Decisions represents each subgoal as a sequence of $N$ 2D waypoints $P = \{p_k\}_{k=1}^N$ projected on the image plane. Each waypoint is annotated with (1) pixel coordinates, (2) a category, (3) an ordering index, and (4) a completion condition, of the subtask. The first two annotations are used to score and rank 3D actions relative to the waypoint, while the latter two determine the transition across subgoals. Further details on the used prompts and examples of generated waypoints are provided in the Appendix.

**Guided action sampling.** To explore a subgoal, Split Decisions modulates policy action sampling using the 2D waypoint information. At each timestep, it samples $M$ candidate actions $A = \{a_i\}_{i=1}^M$ from the policy, ranks them by proximity to the current waypoint, and executes one chosen from the top-ranked set. Unlike reward-shaping methods that guide exploration indirectly by assigning higher rewards to actions closer to the subgoal (Patel et al., 2025), Split Decisions steers exploration directly, avoiding sampling futile actions from low-value regions and thereby achieving greater sample efficiency.

Split Decisions scores 3D actions relative to a 2D waypoint based on their distance, which is inherently ambiguous. To remove the ambiguity, Split Decisions categorizes these waypoints into two scenarios: ones that approach objects (e.g. *grasp the apple* or *put on the plate*), and others that denote intermediate end-effector motion (e.g. *lift the apple* or *move toward the plate*). For object-reaching subgoals, the distance between waypoint $p_k$ and action $a_i$ is computed in 3D: $d(p_k, a_i) = \|\Phi^{-1}(p_k) - a_i^{\text{pos}}\|_2$, where $\Phi^{-1}$ denotes the operator that unprojects the 2D waypoint to 3D, using posed camera information and sensed depth. For intermediate-motion subgoals, the distance is measured in 2D: $d(p_k, a_i) = \|p_k - \Phi(a_i^{\text{pos}})\|_2$, where $\Phi$ denotes the operator that projects the 3D action onto 2D image plane. To prevent collisions, sampled actions too close to any 3D scene point are assigned infinite distance if below a threshold $\tau_c$. Finally, the top-$K$ closest actions are selected, and one is randomly executed to collect interaction data.

After execution, Split Decisions determines whether the current subgoal is achieved based on $d(p_k, a_i)$ and the completion condition provided by the VLM. If $d(p_k, a_i)$ falls below a threshold $\tau_d$ and the condition is satisfied, Split Decisions advances to the next waypoint $p_{k+1}$; otherwise, it continues exploring the current subgoal until the episode ends. In this work, we consider only the grasping condition–whether the end-effector should hold an object–for the current subgoal. For simplification, we detect grasping status using the simulator's built-in detector. Alternative sensing methods, such as motion tracking, could also be applied.

**Training with online or offline RL.** After collecting data with Split Decisions in the exploration phase, we proceed to the learning phase of RL. For online RL, the policy and the critic network are optimized using the PPO objective $L_{PPO}$. These two phases are alternated iteratively. For offline RL, the policy is optimized with DPO objective $L_{DPO}$.

### 4.2 RL FINE-TUNING OPENVLA WITH EXTERNAL GUIDANCE

Although Split Decisions can in principle be applied to both online and offline RL, fine-tuning large vision–language–action models such as OpenVLA remains challenging in practice. In the following paragraphs, we describe these challenges and our proposed solutions.

**Challenge 1: slow action sampling with OpenVLA.** Our approach benefits from evaluating a large set of candidate actions at each step, however, VLAs are autoregressive models: producing a single 6-DoF action requires multiple forward passes. This makes large-scale action sampling prohibitively expensive. To mitigate this, we approximate action sampling by drawing one action $a_0$ from the policy and perturbing its position component with $M$ Gaussian noises $\epsilon_i \sim \mathcal{N}(0, \sigma^2 I)_{i=1}^M$, forming the candidate set $A = \{[a_0^{pos} + \epsilon_i; a_0^{rot}; a_0^{open}]\}_{i=1}^M$, where action $a_0^{pos} \in \mathbb{R}^3$, $a_0^{rot} \in \mathbb{R}^3$ and

$a_0^{open} \in \mathbb{R}$ specify the position, orientation, and gripper opening of action $a_0$, respectively. This technique achieves a practical balance between computational efficiency and sampling diversity.

**Challenge 2: unimodal probability distribution of OpenVLA.** While Gaussian perturbations improve the efficiency of action sampling, many candidate actions fall outside OpenVLA's output distribution. Fig. 3 plots the log likelihood of each action in the candidate set $A$ against its distance from the originally sampled action $a_0$. Open-VLA assigns high confidence only within a narrow region of action space, whereas high-value actions suggested by VLMs often receive near-zero probabilities. This mismatch between the policy's belief and the guided actions makes RL fine-tuning with PPO or DPO ineffective. Since both objectives (Eq. (1), Eq. (2)) optimize actions only relative to the base policy ($\pi_{old}$ or $\pi_{ref}$), training becomes inefficient when action samples are assigned vanishing probabilities.

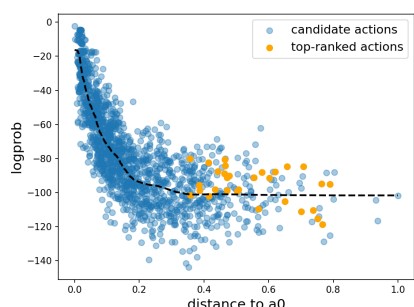

Figure 3: **Log-likelihood vs. distance around a sampled action.** The log likelihood of each action in the candidate set $A$ against its distance from the originally sampled action $a_0$. OpenVLA is overly confident within a narrow region of action space.

To address these challenges, we adapt standard online and offline RL pipelines to OpenVLA. For online RL, we adopt a staged training strategy. In the early stage, Split Decisions guides exploration, and the policy is optimized with the Policy Gradient (PG) objective (Sutton et al., 1998): $L_{PG}(\theta) = -\mathbb{E}[\sum_{t=0}^{T-1} \gamma^t r_t]$. In the later stage, we disable the guidance and continue training with the standard PPO algorithm. For offline RL, we augment the training objective with an additional supervised fine-tuning term $L_{SFT}(\theta) = -\mathbb{E}_{(o,a)\sim\mathcal{D}}[\log \pi_\theta(a|o)]$, ending up with a total loss $L_{ALL} = L_{DPO} + \lambda L_{SFT}$ with $\lambda$ controlling the weighting between preference optimization and imitation learning. Our experiments show that these simple remedies substantially improve sample efficiency.

## 5 EXPERIMENTS

Our experiments are designed to address the following questions: (1) Can Split Decisions enhance sample efficiency of online RL fine-tuning OpenVLA? (2) Can Split Decisions create better datasets for offline RL? (3) Does Split Decisions generalize to different types of manipulation tasks? (4) Which model component is critical for the efficacy of Split Decisions?

### 5.1 BENCHMARK

SimplerEnv (Li et al., 2024b) is a simulation benchmark for robotic manipulation, designed as a digital twin of real-world environments from BridgeData (Walke et al., 2023) and Google Robot (Brohan et al., 2022). It reproduces the same robot arms–WidowX-250 (6-DoF) and Everyday (7-DoF)– along with calibrated control parameters, camera viewpoints, and object properties (geometry, appearance, and physical parameters), to align simulation behavior with real-world performance. Despite providing a unified platform for evaluating manipulation policies, SimplerEnv remains limited in scope, containing only 4 tasks ('Put spoon on towel', 'Put carrot on plate', 'Stack green block on yellow block', and 'Move eggplant into basket') from BridgeData and 5 tasks ('Pick coke can', 'Move near', 'Open drawer', 'Close drawer' and 'Open drawer and place apple') from Google Robot. To increase task diversity and realism, we follow Liu et al. (2025) to create a more challenging BridgeData variant. We place multiple objects–a carrot, a shovel, a cloth and a plate–in a scene, and task the policy to perform 4 manipulation tasks ('Put carrot on plate', 'Put carrot on cloth', 'Put shovel on plate', 'Put shovel on cloth') from the same scene.

We evaluate online RL fine-tuning on our proposed simulation benchmark. For offline RL fine-tuning, we adopt both BridgeData and Google Robot simulation benchmark. Performance is measured by task success rate, defined as the proportion of episodes in which the goal condition is satisfied. Unless otherwise specified, we use the same episode lengths reported by baseline methods.

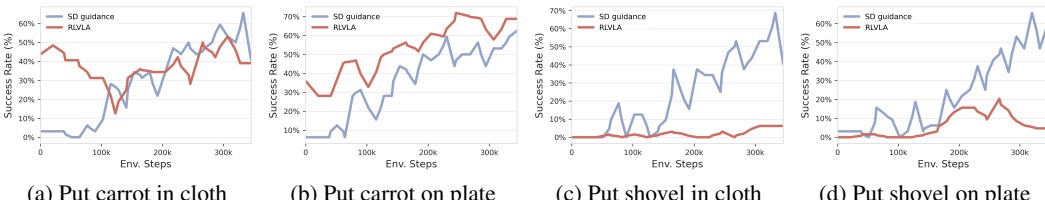

(a) Put carrot in cloth     (b) Put carrot on plate     (c) Put shovel in cloth     (d) Put shovel on plate

Figure 4: **Evaluation on our simulation benchmark using multi-task online RL setting**. Compared against RLVLA (Liu et al., 2025), our Split Decisions enhances the sample efficiency of online RL, enabling the fine-tuned OpenVLA to complete two shovel manipulation tasks.

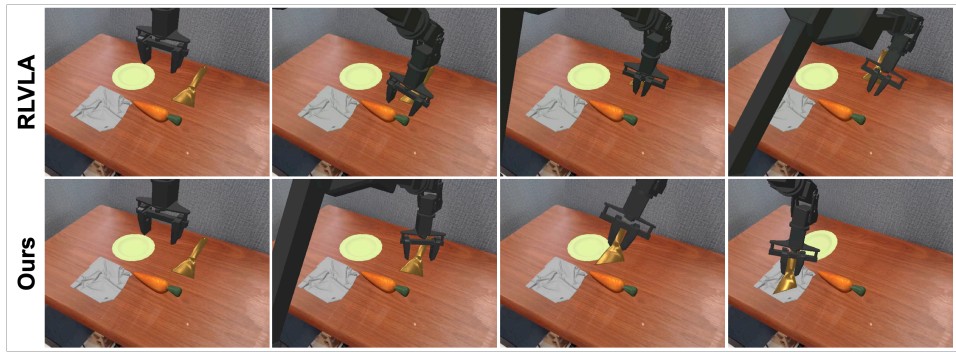

Figure 5: **Visual results of online RL**. From left to right, we present the trajectory rollout during inference. Our method is able to complete the task successfully.

## 5.2 CAN SPLIT DECISIONS ENHANCE ONLINE RL FINE-TUNING OPENVLA?

We evaluate Split Decisions in the online RL setting on our proposed simulation benchmark. We fine-tune a pre-trained policy in a multi-task manner.

**Baselines.** We compare against RLVLA (Liu et al., 2025), a state-of-the-art method that augments OpenVLA with a value estimator and fine-tunes it using PPO. Our setup follows RLVLA exactly, with differences only in the exploration strategy and training objective. During exploration, Split Decisions leverages VLM-guided action sampling, whereas RLVLA samples actions directly from the policy. During training, we collect interaction data under guidance of VLMs and optimize the PG objective $L_{PG}$ in the early stage; we turn off the guidance and optimize the PPO objective $L_{PPO}$ in the later stage. RLVLA relies solely on PPO. In this experiment, we place the four objects at four different positional configurations. We train and test policies on these configurations.

As shown in Fig. 4 and Fig. 5, our Split Decisions enhances the sample efficiency of online RL. Specifically, it enables the fine-tuned OpenVLA to complete two shovel manipulation tasks, achieving around 70% and 60% success rate with 300K environment action steps. In contrast, the policy fine-tuned with RLVLA fails to execute the task.

## 5.3 CAN SPLIT DECISIONS ENHANCE OFFLINE RL DATASET GENERATION?

We evaluate Split Decisions in offline RL settings across two benchmarks: BridgeData simulation and Google Robot simulation, covering both pick-and-place and drawer manipulation tasks.

**Experimental Setup.** We collect offline datasets with Split Decisions and use them to fine-tune OpenVLA across multiple tasks. We compare against: (1) Random sampling baselines using OpenVLA and Octo (Team et al., 2024) for demonstration collection; (2) GRAPE (Zhang et al., 2024), a preference-based RL framework using VLMs for trajectory ranking (BridgeData only). For random sampling and Split Decisions, we fine-tune OpenVLA using a joint objective: $L_{ALL} = L_{DPO} + \lambda L_{SFT}$ with $\lambda = 2.0$.

Table 1: **Offline RL performance on BridgeData simulation.** We evaluate whether Split Decisions can enhance sample efficiency by collecting offline demonstrations and fine-tuning a pre-trained OpenVLA policy. Compared to random sampling (Octo, OpenVLA) and GRAPE, Split Decisions achieves the highest grasp and success rates across all four tasks, yielding a substantial improvement in average performance.

| | Put spoon | | Put carrot | | Stack block | | Move eggplant | | Average | |
|---|---|---|---|---|---|---|---|---|---|---|
| | Grasp | Succ. | Grasp | Succ. | Grasp | Succ. | Grasp | Succ. | Grasp | Succ. |
| before finetuning | 5% | 0% | 15% | 0% | 10% | 5% | 15% | 0% | 11.3% | 1.3% |
| Random sampling (Octo) | **85**% | 45% | 25% | 10% | 55% | 10% | 65% | 45% | 57.5% | 27.5% |
| Random sampling (OpenVLA) | 25% | 20% | 35% | 15% | 40% | 5% | 55% | 20% | 38.8% | 15% |
| GRAPE | 70% | **50**% | 35% | 35% | 50% | 20% | 50% | 5% | 51.3% | 27.5% |
| Split Decisions (ours) | 65% | 35% | **50**% | **45**% | **95**% | **70**% | **100**% | **70**% | **78.8**% | **55**% |

**Results.** On BridgeData simulation, Split Decisions significantly outperforms all baselines, achieving 78.8% grasp rate and 55% success rate, compared to GRAPE's 51.3% and 27.5%, respectively. According to Table 2, the improvement is even more pronounced on Google Robot's drawer manipulation tasks, where Split Decisions achieves a 40.8% performance gain over random sampling baselines. These results demonstrate that Split Decisions provides higher-quality demonstrations for offline RL, generalizing effectively across diverse manipulation tasks from simple pick-and-place to complex drawer operations.

## 5.4 ABLATION STUDIES

We conduct comprehensive ablation studies to evaluate the effectiveness of each model component. All experiments are performed on the BridgeData simulation benchmark across all four tasks under the offline RL setting.

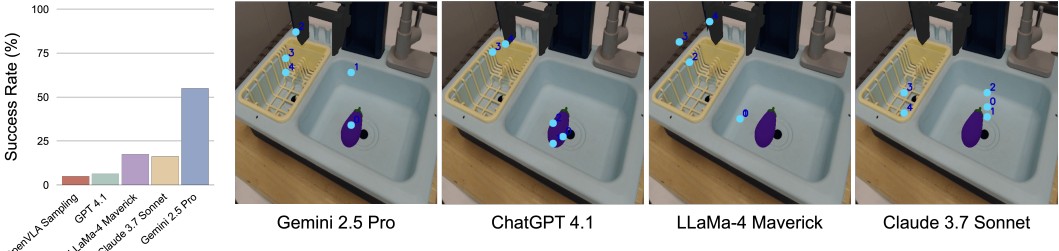

Figure 6: **Generalization across different VLMs.** We evaluate Split Decisions under the multi-task offline RL setting on the BridgeData simulation benchmark, guided by different VLMs. **Left:** Success rates averaged over task variations show that structured subgoals consistently improve performance, with Gemini 2.5 Pro providing the strongest gains. **Right:** Visualizations of waypoint subgoals generated by different VLMs, where numbers indicate their temporal order.

**Choice of VLMs.** We evaluate Split Decisions's generalization capability across different VLMs on the complete BridgeData simulation benchmark. As illustrated in Fig. 6, ChatGPT-4.1 (OpenAI, 2025) demonstrates limited performance with an average success rate of 6.3%. Llama 4 Maverick (AI, 2025) and Claude 3.7 Sonnet (Anthropic, 2025) show moderate improvements, achieving 17.5% and 16.3% average success rates, respectively. Notably, Gemini 2.5 Pro achieves the highest performance with a mean success rate of 55.0%. These substantial performance gaps indicate that stronger vision-language priors directly contribute to higher-quality subgoal generation, particularly in tasks requiring precise spatial reasoning.

**Number of Candidate Actions.** We investigate the impact of varying the number of sampled actions (candidate actions) in Split Decisions across all four tasks in the BridgeData simulation benchmark. As shown in Table 3(a), increasing the number of sampled actions from 100 to 600

yields consistent improvements in both grasp and task success rates. Specifically, grasp rates improve from 54% to 67%, while success rates increase from 50% to 55%. This trend suggests that a larger sampling pool enables the model to explore a more diverse action space, thereby increasing the likelihood of finding optimal trajectories.

**Number of Top-Ranked Actions.** We examine the effect of varying the number of top-ranked ($K$) in Split Decisions on the BridgeData simulation benchmark. As demonstrated in Table 3(b), increasing $K$ from 1 to 20 results in marginal improvements in grasp rates (from 65% to 67%) while achieving a more notable increase in success rates (from 51% to 55%). Further increasing $K$ to 40 improves grasp rates to 70% while maintaining the success rate at 55%. These results indicate that success rates plateau at $K = 20$, suggesting that a moderate candidate pool size optimally balances exploration and exploitation for task completion.

**Variance of Gaussian Noise.** We analyze how the variance of Gaussian noise used for action augmentation affects Split Decisions's performance on the BridgeData simulation benchmark. As presented in Table 3(c), increasing the variance from 0.01 to 0.04 leads to substantial improvements, with success rates rising from 49% to 55% and grasp rates improving slightly from 66% to 67%. Further increasing the variance to 0.10 maintains these performance levels without additional gains. This pattern indicates that moderate noise levels effectively expand the exploration radius, enabling the sampler to propose action candidates that better align with target waypoints, while excessive noise provides no additional benefit.

| | Close Drawer | | | Open Drawer | | | Average |
|---|---|---|---|---|---|---|---|
| | Bottom | Middle | Top | Bottom | Middle | Top | |
| before finetuning | 11.1% | 0% | 11.1% | 0% | 0% | 22.2% | 7.4% |
| Random sampling (Octo) | 0% | 0% | 1% | 0% | 0% | 0% | 1.9% |
| Random sampling (OpenVLA) | 55.6% | 22.2% | 55.6% | 11.1% | 22.2% | 33.3% | 33.3% |
| Split Decisions (ours) | **88.9**% | **88.9**% | **88.9**% | **55.6**% | **44.5**% | **77.8**% | **74.1**% |

Table 2: **Generalization to drawer manipulation tasks.** We evaluate Split Decisions in a multi-task offline RL setting on the Google Robot benchmark with 'Open Drawer' and 'Close Drawer' tasks. Split Decisions achieves substantially higher success rates across all drawer positions, yielding an average of 74.1% versus 33.3% for the strongest baseline.

| (a) Candidate Actions | | | (b) Top-K Selection | | | (c) Action Variance | | |
|---|---|---|---|---|---|---|---|---|
| # Actions | Grasp | Success | K | Grasp | Success | $\sigma^2$ | Grasp | Success |
| 100 | 54% | 50% | 1 | 65% | 51% | 0.01 | 66% | 49% |
| 300 | 56% | 52% | 20 | 67% | 55% | 0.04 | 67% | 55% |
| 600 | 67% | 55% | 40 | 70% | 55% | 0.10 | 67% | 55% |

Table 3: **Hyperparameter ablation on success rate.** We report grasp and task success while varying the number of candidate actions, top-$K$ selection, and action variance.

# 6 SUMMARY

We present Split Decisions, an exploration framework that leverages vision-language models (VLMs) to provide structured 2D waypoint subgoals for guiding vision-language-action models (VLAs). By introducing this high-level semantic guidance, Split Decisions reduces wasted interactions, enhances the quality of collected trajectories, and improves the efficiency of policy adaptation in both online and offline reinforcement learning. While the framework demonstrates strong effectiveness, its current reliance on 2D guidance constrains precise 3D spatial reasoning and complex manipulation. Future work should focus on extending VLM guidance to 3D-aware subgoals, enabling richer spatial constraints and strengthening applicability in real-world robotic settings.

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

# Table of Contents

## A  DECLARATION OF LLM USAGE

We used large language models (LLMs) to assist with writing refinement (e.g., grammar, spelling, word choice), to generate waypoint trajectories for our method, and to support code implementation.

## B  FULL WORKFLOW OF SPLIT DECISIONS GUIDANCE

Our Split Decisions framework leverages vision-language models (VLMs) to guide exploration by generating semantic subgoals. Instead of relying solely on random action sampling, the policy is directed by VLM-provided priors, which help identify meaningful waypoints and reduce wasted interactions. This guidance integrates seamlessly into the pipeline shown in Algorithm 1, enabling more sample-efficient and goal-directed rollouts.

---

**Algorithm 1:** VLM-Guided Exploration

---

**Inputs:** Task instruction $l$, episode horizon $T$, # of waypoints $N$, # of sampled actions $M$, # of top-ranked actions $K$, VLM, policy $\pi_\theta$, 2D-to-3D unprojection $\Phi^{-1}$, 3D-to-2D projection $\Phi$, variance $\sigma$, and environment env

**Output:** Replay buffer $B$

---

Initiate replay buffer $B = \{\}$
Initial observation $o = $ env.get_obs()
Generate 2D waypoint trajectories $P = \{p_k\}_{k=1}^N \leftarrow \text{VLM}(o, l)$
Waypoint index $k \leftarrow 1$
**for** $t = 1$ **to** $T$ **do**
    Sample a random action $a_0 = [a_0^{pos}; a_0^{rot}; a_0^{open}] \sim \pi_\theta(o, l)$
    Sample random noise $\{\epsilon_i \sim \mathcal{N}(0, \sigma)\}_{i=1}^M$
    Create a set of random actions $A = \{[a_0^{pos} + \epsilon_i; a_0^{rot}; a_0^{open}]\}_{i=1}^M$

    **if** $p_k$.category *is object-reaching* **then**
        Calculate distance in 3D: $\{\|\Phi^{-1}(p_k) - a_i^{\text{pos}}\|_2\}_{i=1}^M$
    **else**
        Calculate distance in 2D: $\{\|p_k - \Phi(a_i^{\text{pos}})\|_2\}_{i=1}^M$

    Select top-ranked action candidates $\bar{A}$
    Randomly sample one action from $a \sim \text{Uniform}(\bar{A})$
    Execute the action $o, r = $ env.step(a)
    Update the replay buffer $B \leftarrow B \cup \{(o, a, r)\}$
    **if** $p_k$.completed *is True* **then**
        Proceed to the next waypoint $k \leftarrow k + 1$

---

## C    ONLINE RL CONFIGURATION WITH SPLIT DECISIONS

We train and evaluate Split Decisions in a tabletop scene with 32 parallel environments containing two manipulable objects and two receptacles, instantiated with four goal instructions—*put carrot in cloth*, *put carrot in yellow plate*, *put kitchen shovel in cloth*, and *put kitchen shovel in yellow plate*. Object and receptacle poses are randomized across four spawn locations during both training and inference to assess robustness to spatial variation. The policy is warm via Policy Gradient on 50 demonstrations, capping each environment at 3 trajectories to promote scene diversity; when full task completions are sparse, we relax the collection criterion to grasp success to ensure coverage of contact primitives. In each step, we sample 70 candidate actions by gaussian perturbations around the policy action, score candidates with the VLM, and execute from the top–$K$ set (defaults $K{=}10$, $\sigma{=}0.15$). The training is carried out for 350,000 steps with a learning rate $1 \times 10^{-4}$.

## D    OFFLINE RL ROLLOUT COLLECTION PERFORMANCE

Table 4 shows that Split Decisions-guided collection outperforms random sampling with OpenVLA, yielding higher grasp and task success across SimplerEnv and Google Robot. This highlights that semantic priors improve sample efficiency and produce higher-quality offline datasets for stronger downstream RL performance.

| Task | OpenVLA sampling | | Split Decisions / OpenVLA | |
|---|---|---|---|---|
| | Grasp | Success | Grasp | Success |
| put the spoon on the towel | 25% | 20% | 90% | 65% |
| put carrot on plate | 35% | 20% | 95% | 55% |
| stack the green block on the yellow block | 40% | 5% | 90% | 70% |
| put eggplant into yellow basket | 25% | 25% | 100% | 85% |
| close drawer (bottom) | – | 55.6% | – | **88.9%** |
| close drawer (middle) | – | 22.2% | – | **77.8%** |
| close drawer (top) | – | 55.6% | – | **100.0%** |
| open drawer (bottom) | – | 11.1% | – | **55.6%** |
| open drawer (middle) | – | 44.4% | – | **55.6%** |
| open drawer (top) | – | 88.9% | – | **100%** |
| **Average (SimplerEnv)** | 31.3% | 17.5% | **93.8%** | **68.8%** |
| **Average (Google Robot)** | – | 46.2% | – | **79.6%** |

Table 4: **Offline RL rollout evaluation across multiple benchmarks.** We report grasp and task success percentages for OpenVLA random sampling versus our Split Decisions-guided data collection. Results include four in-domain tasks from SimplerEnv as well as 'Open Drawer' and 'Close Drawer' tasks from the Google Robot benchmark. Using Split Decisions to collect offline rollouts consistently improves data quality, yielding 81.3% success on SimplerEnv and 74.1% on Google Robot drawer tasks after fine-tuning.

## E    OFFLINE REINFORCEMENT LEARNING TRAINING CONFIGURATION

This configuration demonstrates a hybrid training approach for offline reinforcement learning that combines Direct Preference Optimization (DPO) with Supervised Fine-Tuning (SFT) for a Vision-Language-Action (VLA) model. The base model `openvla-7b` serves as both the policy network (`vla_path`) and reference model (`ref_path`) for DPO training. The training utilizes the `sd_bridge` dataset with a conservative batch size of 1 and window size of 1, running for 120,000 steps with learning rate decay from $3 \times 10^{-4}$ to $1 \times 10^{-8}$.

The key innovation lies in the mixed loss formulation, where `use_mixed_loss` enables simultaneous optimization of both DPO and SFT objectives. The `tpo_beta` parameter of 0.05 controls the KL divergence penalty in the DPO loss function $\mathcal{L}_{\text{DPO}} = -\log \sigma \left( \beta \log \frac{\pi_\theta(y_w|x)}{\pi_{\text{ref}}(y_w|x)} - \beta \log \frac{\pi_\theta(y_l|x)}{\pi_{\text{ref}}(y_l|x)} \right)$,

balancing exploration versus staying close to the reference policy. The `sft_loss_weight` of 2.0 with `dynamic_sft_weight` enabled allows the model to adaptively balance between behavior cloning from demonstrations and preference-based optimization. This is particularly crucial in offline RL settings where the model must learn from fixed datasets without environmental interaction. The configuration employs LoRA (Low-Rank Adaptation) with rank 64 for parameter-efficient fine-tuning, making it suitable for resource-constrained deployments while maintaining model expressiveness. The `length_normalize` flag ensures fair comparison between trajectories of different lengths, addressing a common challenge in offline RL where demonstration quality varies with trajectory length.

## F  UNIFIED PROMPT DESIGN FOR VLMS

We carefully design a text prompt that encourage VLMs to generate consistent and accurate waypoint trajectories. For most of our experiments, we use **Gemini 2.5 Pro** as the VLM, which generates the most spatially accurate and semantically correct waypoints for the manipulation task.

To evaluate the robustness and generality of our method across different VLMs, we compare **Gemini 2.5 Pro** against 3 other VLMs: **ChatGPT 4.1**, **Claude 3.7 Sonnet**, and **LLaMA-4 Maverick**. All four models were queried using the same prompt template, enabling a controlled comparison of their spatial reasoning and instruction grounding capabilities.

---

### VLM prompt template used across all models

You are given a top-down RGB image, where the origin (0,0) is at the top-left corner, x increases to the right, and y increases downward. Coordinates should be reported as integer pairs in [y, x] format, normalized to the 0–1000 range. This is very important that the point you provide be precisely and completely correct.

Point to where a robot end-effector would TASK_DESCRIPTION, and create a trajectory of 5 points connecting them. Determine the correct optimal grasping point on the object by inferring it from the object's shape (i.e., the most stable and physically accessible location for a secure grasp, it should lie exactly on the visible part of the target object surface) and the best placement point within the target area (specifically, the center of the plane). Plan a smooth and collision-free trajectory from the robot gripper position to the object to ensure safe motion through all waypoints. For pick-up or place tasks, begin the approach by first raising the gripper higher than the object height to maintain full clearance from the desktop and plan waypoints to move toward the target. Additionally, evaluate whether a particular point is intended for a subtask switch based on whether this subtask requires the robot to grasp the object. For each point, include an extra boolean field "grasp_subtask_switch" set to true if this subtask requires a grasp action to interact with the object and trigger a subtask transition (grasping is needed), and false otherwise.

Additionally, propose keypoints that are reasonably and uniformly spaced along the trajectory to ensure a fixed and consistent interval between each point. For any contact-driven action (e.g., push/pull/open/close), infer the object's permissible motion from visible geometry and affordances point, and constrain the end-effector path to lie on the axis for sliding with the motion direction consistent with the task verb(e.g. open/close).

For drawer tasks, please use visual and occlusion reasoning, first resolve the correct drawer by matching the action verb to the object's current state (for example, closed or opened of the drawer). Infer a stable contact point that is reachable and clearly graspable on the visible surface of the correct object and a smooth, collision-free 2D trajectory, ensuring each waypoint is unambiguous in the image (clearly on-object, occlusions, and background). After grasping drawer, infer the correct sliding direction based on the task verb: move outward along the drawer surface normal for open, or inward along the drawer surface normal for close.

Drawer target resolution (level + occlusion reasoning). When the task verb specifies a drawer level, first rank all detected drawer fronts/handles by their handle (or panel-center) y-coordinate from top to bottom to map to levels. Then, always choose a grasp waypoint on the same horizontal side of the drawer as the gripper's current position to minimize travel distance and

---

avoid crossing over the object. (For example, if the gripper is located on the right-hand side of the image, select the grasp point on the right-hand side of the drawer; if the gripper is on the left, select the left-hand side.)

Also, ensure the overall trajectory is smooth, reasonable, avoids any potential collisions and maintains a sufficient clearance and space from irrelevant objects such as the desktop surface. For instruction related to pick-up or placing object, must first raise the gripper even higher than usual while approaching the object to ensure full clearance from the desktop.

Label the points sequentially with integers from "0" (start point) to "n" (final point). Each point should include the corresponding task details and an indication if the robot reaches the corresponding object.

Note: The "approach_object" field indicates whether the robot's end-effector will come into contact with the object.

Provide the answer strictly in JSON format as follows, do not contain any other words or symbols: `[{"point": <point in [y, x] format normalized to 0-1000>, "label": <label>, "task": <task description>, "approach_object": <True/False>, "grasp_subtask_switch": <True/False>}, ...]`.

Before finalizing each point, perform a detailed chain-of-thought analysis. Divide your analysis into two cases:

For points where approach_object is set to true:

Ensure that the point lies on the visible, touchable portion of the target object on the 2D image, clearly on the object itself and not on the gripper or any irrelevant object like the desktop surface. If grasp_subtask_switch is true, then approach_object must also be true. Confirm that the point is not at or near the edge of the object; it should be selected from a stable, central region of the object's surface for a balanced and secure grasp. If the motion involves interacting with the object, double-check that the keypoint is precisely on the object and not on the desktop or any other area. If the analysis indicates that the point is occluded, obstructed, too close to the object's edge, or lacks sufficient clearance, determine whether the point is obstructed or unstable, and if so, propose a new, unobstructed, and stable point on the object.

For points where approach_object is set to false:

Ensure that the point is sufficiently separated from the target object and any irrelevant items (e.g., the desktop) to maintain a clear, unambiguous trajectory key point.

Verify that the selected point provides a clear, unobstructed trajectory key point without causing any collisions or interference with extraneous objects.

You must output only the final answer as valid JSON array and nothing else. Do not include any markdown formatting, no code block, no explanation, no additional text. Just return the raw JSON array. Any non-JSON characters will break the system.

