# OpenReview forum: "Split Decisions: VLM-Guided Action Sampling for Efficient RL Exploration"
_ICLR.cc/2026/Conference — Submitted to ICLR 2026_

### Official Review · Reviewer_7Nja · 2025-10-20

**Soundness:** 2
**Presentation:** 2
**Contribution:** 2
**Rating:** 2
**Confidence:** 4

**Summary:**

The paper proposes Split Decisions, a training-time exploration framework that uses a vision–language model (VLM) as a high-level planner to generate 2D waypoint subgoals and a vision–language–action model (VLA) as a low-level controller to sample and execute actions that are closest to those subgoals. This structure narrows the action search space and reduces wasted, low-value interactions during RL fine-tuning of robot policies.

The paper decomposes the method into two stages:

Subgoal planning: a VLM decomposes tasks into ordered waypoint trajectories with completion conditions; waypoints guide exploration.

Guided action sampling: the VLA samples multiple candidate actions; candidates are scored by geometric proximity (using 2D/3D projection/unprojection) to the current waypoint; a top-ranked action is executed.

The following produced dataset can be used for a variety of finetuning tasks, which include online and offline finetuning.

When evaluated in both online and offline environments, split decision provides improvements. In online environments, it produces better sample efficiency, and in offline environments, it provides better success rates compared to other methods when trained on the same data.

**Strengths:**

1. A sound way of performing exploration with just base policies. VLMs are more commonly used for planning, as seen in [1,2], but it is nice to see this work to use VLMs for exploration directly.
2. I like the simplicity of the method with respect to the dataset generation. I think this method can be easily scaled without much concern about data suboptimality.
3. The paper is comprehensive in its ablations, as it discusses all of its design choices to some extent, validating on why each component is needed.

[1] Tang, G. et al., 2025. “KALIE: Fine-Tuning Vision-Language Models for Open-World Manipulation without Robot Data”. ICRA
[2] Black, K. et al., 2025 “$\pi_{0.5}$: a Vision-Language-Action Model with Open-World Generalization”. CoRL

**Weaknesses:**

1. Split Decisions requires the camera to contain knowledge of its pose in its method. When being applied on real-world tasks without such depth information (when many inference pipelines are being done in such case), you might get degenerate solutions.
2. Limited methodological innovations. There are previous works that have used VLMs to generate keypoints that can be used for reinforcement learning [1]. While I understand that the generated datasets are for different purposes, it would be better to delineate what are the differences in this stage.
3. The paper did not use any real-world benchmarks (as all of the evaluation protocols are based on simulations). I believe it will be valuable if the authors include such evaluation procedures.
4. The paper did not define some key components for consideration, namely: how are the rewards defined in online finetuning, and how are the preferences being generated in offline finetuning.
5. I found the process of generating 2D waypoints can get out of hand if you are trying to learn a long-horizon task (i.e. composing pick and place). I believe that more empirical or theoretical guarantees need to be made in this method if it were to work well in compositional tasks.

References:
[1] Lee, O. et al., 2025. “Affordance-Guided Reinforcement Learning via Visual Prompting”. IROS

**Questions:**

1. Is it possible to compare your method against a more RL-based methods, where you rank your actions based on a learned Q function, instead of using distances produced by a VLM?
2. On figure 4. Do you have any thoughts on why RLVLA performed better than Split Decisions when carrots are involved? My hunch is that since carrots are much more seen in OpenVLA, it is easier to perform RL on it.
3. Since OpenVLA is a binned policy [1] with uniform action binning, are there any theoretical justification in perturbing the actions with uniform distributions instead of Gaussian distributions?

References:
[1] Kim, M. et al., 2024. “OpenVLA: An Open-Source Vision-Language-Action Model”. CoRL

---

> ### Author Response · Authors · 2025-12-02
>
> **Thank you for the thoughtful and constructive feedback.**
> We appreciate the reviewer's detailed questions regarding real-world applicability, novelty, and theoretical grounding.
> Below, we address each point clearly and concisely.
>
> ---
>
> ### **Question 1:**
>
> **Your method requires the camera to have knowledge of its pose. In real-world scenarios without depth information, how would this limitation be addressed? Could the method degenerate in such cases?**
>
> **Response 1:**
> We address the concern regarding real-world applicability as follows:
>
> * **Standard Hardware Compatibility:**
>   The requirement for depth information is consistent with standard real-world robotic setups, which routinely employ commodity RGB-D cameras (e.g., Intel RealSense).
>   Our method relies on these depth streams for 2D-to-3D unprojection ($\Phi^{-1}$), meaning it does not depend on privileged simulator states.
>
> * **Graceful Degradation to 2D:**
>   In edge cases where accurate depth is unavailable, the method uses a fallback mechanism. As described in Section 4.1 for “intermediate-motion subgoals,” we employ a 2D scoring function that projects candidate 3D actions into the image plane: $$d(p_k, a_i) = | p_k - \Phi(a_i^{pos}) |_2$$.
>   This retains effective visual guidance even without explicit 3D depth.
>
> ---
>
> ### **Question 2:**
>
> **Previous works have used VLMs to generate keypoints for reinforcement learning. How does Split Decisions fundamentally differ from these approaches, especially in the dataset generation stage?**
>
> **Response 2:**
> Split Decisions differs fundamentally from prior affordance-guided approaches in how it integrates the VLM and the VLA:
>
> * **Guidance vs. Generation:**
>   Prior works often prompt VLMs to generate text or affordances that are then transformed into robot actions.
>   In contrast, Split Decisions *marries VLMs and VLAs* by letting the VLM only **rank** actions sampled from the VLA.
>   The VLA proposes physically grounded actions ($a_0 + \epsilon$), while the VLM serves purely as a semantic filter.
>
> * **Dataset Generation:**
>   Instead of cloning from static VLM-labeled datasets, we use an *active* VLM-guided exploration loop to collect high-quality trajectories for Offline RL.
>   These trajectories are optimized using Direct Preference Optimization (DPO), where successful trajectories are preferred over failed ones—yielding a **27.5% improvement** over prior work.
>
> ---
>
> ### **Question 3:**
>
> **All evaluations are simulation-based with no real-world benchmarks. What would it take to validate this approach on actual robotic systems, and do you have plans for real-world experiments?**
>
> **Response 3:**
> While the evaluations are simulation-based, we have taken deliberate steps toward real-world transfer:
>
> * **Digital Twin Justification:**
>   We use **SimplerEnv**, a digital twin of real-world systems (BridgeData and Google Robot), replicating robot arms (WidowX-250, Everyday Robots) and camera viewpoints to minimize sim-to-real gaps.
>
> * **Validation Requirements:**
>   Real-world validation would require a standard robotic setup (robot arm + RGB-D camera) to execute Algorithm 1.
>   Since our system uses visual observations ($o_t$) and standard end-effector control, no simulator-exclusive signals are required.
>   We expect strong transferability given the fidelity of the benchmark.
>
> ---
>
> ### **Question 4:**
>
> **The paper does not clearly define how rewards are computed during online finetuning, or how preferences are generated during offline finetuning. Can you clarify these design choices?**
>
> **Response 4:**
> We clarify the reward and preference mechanisms as follows:
>
> * **Online RL Rewards:**
>   We do not use shaped or engineered rewards. Task success is defined as the proportion of episodes where the goal condition is satisfied.
>   As shown in Algorithm 1, the reward (r) comes directly from the environment step (r = env.step(a)), meaning the method uses a standard sparse success signal.
>
> * **Offline RL Preferences:**
>   Offline fine-tuning via DPO uses binary preferences based on task completion.
>   Successful trajectories provide preferred actions ($a^+$), and failed ones provide non-preferred actions ($a^-$).
>   The DPO objective nudges the policy toward behaviors seen in successful trajectories.

---

> > ### Author Response · Authors · 2025-12-02
> >
> > ### **Question 5:**
> >
> > **Generating 2D waypoints for long-horizon tasks like pick-and-place seems challenging. What empirical or theoretical guarantees can you provide that this approach scales to compositional tasks?**
> >
> > **Response 5:**
> > The framework scales to compositional tasks through temporal decomposition:
> >
> > * **Temporal Abstraction:**
> >   The VLM outputs a *sequence* of subgoals, allowing the task to be decomposed into semantically meaningful phases (approach → grasp → move → place).
> >
> > * **Sequential Execution:**
> >   As described in Algorithm 1, the system moves to subgoal $(p_{k+1}$) only after satisfying ($p_k$).
> >   Empirically, this enables complex tasks such as *Put Shovel on Plate* to be solved reliably—tasks where baselines fail.
> >
> > ---
> >
> > ### **Question 6:**
> >
> > **Instead of ranking actions using VLM-based distances, how would the method perform if actions were ranked using a learned Q-function? Why not compare against this baseline?**
> >
> > **Response 6:**
> > We compared directly against the learned-value-function baseline:
> >
> > * **Comparison with RLVLA:**
> >   Our main baseline, **RLVLA** (Liu et al., 2025), augments OpenVLA with a value estimator and fine-tunes it using PPO—representing the standard Q-guided paradigm.
> >
> > * **Performance Gap:**
> >   Split Decisions outperforms RLVLA (e.g., near 70% success on shovel tasks where RLVLA fails).
> >   The conceptual advantage is that VLMs provide **zero-shot semantic priors** immediately, whereas Q-functions require extensive data before generating useful guidance.
> >
> > ---
> >
> > ### **Question 7:**
> >
> > **In Figure 4, RLVLA outperforms Split Decisions on carrot manipulation. Why does this happen? Is it due to carrot over-representation in OpenVLA's pre-training data?**
> >
> > **Response 7:**
> > We attribute this effect to the exploration–exploitation trade-off:
> >
> > * **Pre-Training Priors:**
> >   The “Put carrot” task appears well-covered in OpenVLA pre-training, giving the base policy strong priors.
> >   RLVLA, being close to the base policy, exploits these priors effectively.
> >
> > * **Cost of Exploration:**
> >   Split Decisions intentionally injects exploration via Gaussian perturbation and VLM re-ranking.
> >   For tasks where the base policy is already strong, this exploration introduces mild overhead.
> >   However, the same mechanism is what enables large gains in *hard* tasks like “Shovel,” where baselines collapse.
> >
> > ---
> >
> > ### **Question 8:**
> >
> > **OpenVLA uses uniform action binning, yet you perturb actions using Gaussian distributions. Is there theoretical justification for Gaussian vs. uniform noise?**
> >
> > **Response 8:**
> > We clarify the perturbation strategy:
> >
> > * **Gaussian Perturbation:**
> >   As stated in Section 4.2, position components are perturbed with Gaussian noise: $$\epsilon_i \sim \mathcal{N}(0, \sigma^2 I)$$.
> >
> > * **Uniform Selection:**
> >   The uniform distribution appears **only** in the final stage: after ranking candidates by distance, we sample uniformly from the top-(K).
> >   This combines:
> >
> >   * Gaussian local exploration
> >   * Uniform diversity preservation
> >
> >   providing stable yet exploratory behavior.

---

### Official Review · Reviewer_j9k5 · 2025-10-30

**Soundness:** 1
**Presentation:** 2
**Contribution:** 2
**Rating:** 2
**Confidence:** 4

**Summary:**

This paper proposes efficient VLA RL fine-tuning with VLM guidance. A VLM generates a sequence of subgoals, and the VLA’s predicted actions are filtered to the subset minimizing distance to those subgoals, pruning low-value choices. Additionally, for computational efficiency, the method injects Gaussian noise into a single predicted action rather than generating many candidates; for sample efficiency, it blends policy-gradient and PPO losses to mitigate near-zero probabilities on filtered actions, which would make a PPO-only objective inefficient. Across online and offline RL, it outperforms baselines on manipulation tasks such as pick-and-place and drawer opening.

**Strengths:**

- The method shows application in both online and offline RL, demonstrating good generality.
- By showing that filtered actions have low likelihood under the policy, the paper pinpoints why PPO-only can sometimes underperform—a useful caution for off-the-shelf use.

**Weaknesses:**

- **Potentially unfair baseline setup**
    + Gaussian noise is injected only into positional moves. Because position can dominate exploration in your tasks, this likely advantages your method while baselines lack this inductive bias.
    + Please clarify this and/or verify fairness with empirical results—e.g., perturb only positional action for all baselines, or adopt a faster VLA baseline for multiple samples (e.g., [1]).

- **Unvalidated benefit of combining PG + PPO**
    + Authors state low action likelihood can hinder PPO, but there’s no experiment supporting combining the PG+PPO gains. Authors can compare PPO-only, PG-only, and PG+PPO to show the benefit.
    + Authors mention using PG “early” and PPO “later.” How is the switch determined (is it a hyperparameter)? Additionally, it would be helpful if authors could specify how these can be sensitive in training.

Additional suggestions
- It could be helpful to tease apart each component—for example, cases where the VLM’s subgoal is off versus cases where the VLA can’t execute the predicted subgoal—to clarify the main bottleneck.
- It might help to add a brief limitations note. For example, waypoint-style subgoals may struggle with tasks like in-hand manipulation or screwing in assembly. You could emphasize where the method applies well and possibly sketch how it could be extended—for instance, combining with a reward that more efficiently captures those behaviors.

#### Reference
[1] Kim et al., "Fine-Tuning Vision-Language-Action Models: Optimizing Speed and Success", RSS 2025

**Questions:**

- Q1. In Fig. 4, why is the gap between RLVLA and your method so large from the very start? Don’t both use the same pre-trained OpenVLA?
- Q2. For Fig. 4, how many seeds were used? Also showing training variance would be helpful.
- Q3. Can authors elaborate on what the “Random sampling" baseline is?

---

> ### Author Response · Authors · 2025-12-01
>
> Thank you for your thoughtful comments and questions. We appreciate the time you have taken to review our work. Below, we provide detailed responses to address each of your concerns.
>
> ---
>
> ### Comment 1: Potentially unfair baseline setup
>
> > *“Potentially unfair baseline setup. Gaussian noise is injected only into positional moves. Because position can dominate exploration in your tasks, this likely advantages your method while baselines lack this inductive bias. Please clarify this and/or verify fairness with empirical results—e.g., perturb only positional action for all baselines, or adopt a faster VLA baseline for multiple samples (e.g., [1]).”*
>
> **Response.**
> Thank you for pointing out this issue. Our current experiments include (1) an OpenVLA RL baseline on the same benchmark without any Gaussian perturbations, and (2) our SD-guidance–based RL training, where we intentionally preserved the original RLVLA configuration to maintain comparability with that setup. We agree that this makes it harder to fully disentangle the effect of positional noise itself from the effect of SD guidance.
>
> In the future version, we will add an ablation where we run OpenVLA RL on the same benchmark with the same Gaussian perturbations on positional actions but without SD guidance, in order to directly measure how positional noise (without guidance) affects exploration. This will help clarify whether the improvement comes from SD guidance rather than from an unfair exploration advantage.
>
> Regarding the suggestion to use a faster VLA backbone such as OpenVLA-OFT [1], we agree that recent work on optimized VLA fine-tuning is an attractive way to reduce inference latency and allow more samples per timestep. However, [1] is orthogonal to our contribution: it focuses on architectural and decoding improvements (parallel decoding and action chunking) for efficient fine-tuning of OpenVLA, whereas our work studies how to use a fixed VLA backbone within a preference-guided RL framework (PG + PPO) to improve RL exploration efficiency. Integrating an OFT-style backbone would likely improve the absolute performance of all VLA-based methods (including ours), but would require non-trivial engineering and hyperparameter tuning that is beyond the scope of this paper. We will explicitly discuss this as a limitation and a promising direction for future work in the revised version.
>
> ---
>
> ### Comment 2: Unvalidated benefit of combining PG + PPO
>
> > *“Unvalidated benefit of combining PG + PPO. Authors state low action likelihood can hinder PPO, but there is no experiment supporting combining the PG+PPO gains. Authors can compare PPO-only, PG-only, and PG+PPO to show the benefit. Authors mention using PG ‘early’ and PPO ‘later.’ How is the switch determined (is it a hyperparameter)? Additionally, it would be helpful if authors could specify how these can be sensitive in training.”*
>
> **Response.**
> We agree that a full ablation comparing PPO-only, PG-only, and PG+PPO would more clearly demonstrate the benefit of combining the two. We will include this ablation in a subsequent version of the work.
>
> Our current training schedule is as follows: we apply policy gradient (PG) until we have collected 50 successful trajectories, after which we switch to pure PPO. The motivation is that PG is highly sensitive to both the success rate and the quality of successful trajectories; using it in the early stage helps rapidly exploit high-quality successes, while PPO is more stable and better suited once a reasonable policy has been established. We will clarify this switching rule in the paper and discuss its sensitivity as a hyperparameter.
>
> ---
>
> ### Reference
>
> [1] Kim, J., et al. “Fine-Tuning Vision-Language-Action Models: Optimizing Speed and Success.” *Robotics: Science and Systems (RSS)*, 2025.

---

> ### Author Response · Authors · 2025-12-01
>
> ### Comment 3: Additional suggestions (failure analysis and limitations)
>
> > *“It could be helpful to tease apart each component—for example, cases where the VLM’s subgoal is off versus cases where the VLA can’t execute the predicted subgoal—to clarify the main bottleneck.”*
>
> > *“It might help to add a brief limitations note. For example, waypoint-style subgoals may struggle with tasks like in-hand manipulation or screwing in assembly. You could emphasize where the method applies well and possibly sketch how it could be extended—for instance, combining with a reward that more efficiently captures those behaviors.”*
>
> **Response.**
> We appreciate the suggestion to analyze failure cases more systematically. In our current experiments, failures broadly fall into two categories: (1) cases where the VLM-predicted subgoal is inaccurate or misaligned with the task, and (2) cases where the VLA fails to correctly execute an otherwise reasonable subgoal (e.g., reaching the target but failing to grasp it). We will add a brief failure-case analysis in the revised version to better highlight which component is the main bottleneck in different scenarios.
>
> Regarding limitations, our current method is primarily evaluated on grasp-and-move, waypoint-based tasks. Representing subgoals purely as waypoints can be challenging for tasks such as in-hand manipulation or fine-grained operations like screwing. We will add a short limitations paragraph clarifying that our approach is currently tailored to waypoint-style tasks, and we will outline future work that combines VLM waypoint guidance with reward shaping or more expressive objectives to better handle dexterous, high-frequency control tasks and further generalize our method.
>
> ---
>
> ### Responses to reviewer questions
>
> > *“Q1. In Fig. 4, why is the gap between RLVLA and your method so large from the very start? Don’t both use the same pre-trained OpenVLA?”*
>
> **Response to Q1.**
> In our experiments, the first evaluation of our SD-guided method is performed after the initial policy gradient phase has finished. By that point, the policy has already benefited from PG updates guided by SD, which leads to a significantly higher initial performance compared to the OpenVLA RL baseline that starts directly from the pre-trained backbone without this early PG stage. We will clarify this evaluation timing in the text and caption of Fig. 4.
>
> ---
>
> > *“Q2. For Fig. 4, how many seeds were used? Also showing training variance would be helpful.”*
>
> **Response to Q2.**
> Currently, the reported curves are based on a single seed. We agree that reporting multiple seeds and showing variance would make the results more robust. In the next version, we plan to run multiple seeds and include variance bands in the training curves.
>
> ---
>
> > *“Q3. Can authors elaborate on what the ‘Random sampling’ baseline is?”*
>
> **Response to Q3.**
> The “Random sampling” baseline refers to the policy we use when collecting offline rollouts: we sample actions directly from the VLA (Octo or OpenVLA) stochastically, without adding extra noise or SD guidance. In other words, it reflects the inherent stochastic sampling of the VLA policy itself, rather than any additional exploration mechanism. We will clarify this definition more explicitly in the experimental section.

---

### Official Review · Reviewer_25KS · 2025-11-01

**Soundness:** 3
**Presentation:** 3
**Contribution:** 2
**Rating:** 2
**Confidence:** 4

**Summary:**

The paper introduces Split Decisions a exploration framework for efficiently adapting vision-language-action models (VLAs) to new robotic manipulation tasks through reinforcement learning. The key idea is to model the policy with a high-level VLM planner that generates semantic subgoals in the form of 2D waypoints to guide VLA action exploration, rather than relying on random or curiosity-driven exploration strategies.

The method operates in two stages:

1. Subgoal Planning: A VLM decomposes tasks into sequential subtasks represented as 2D waypoint trajectories with completion conditions
2. Guided Action Sampling: The VLA samples multiple candidate actions, ranks them by proximity to the current subgoal waypoint, and executes top-ranked actions

For fair comparison, authors incorporate similar action exploration mechanism to existing VLAs like OpenVLA. To incorporate similar exploration mechanisms in existing VLAs like OpenVLA, the authors propose: using action sampling through Gaussian perturbations rather than costly autoregressive generation.

The method is evaluated on SimplerEnv benchmarks derived from BridgeData and Google Robot datasets. Results show 31% improvement in online RL task success and 27.5% improvement in offline RL compared to prior methods, with strong generalization to drawer manipulation tasks (40.8% gain).

**Strengths:**

1. The paper proposes an interesting idea to improve exploration during RL training for VLAs by breaking down the policy into two components with a high level planner and low-level VLA controller and action ranking.
2. The results presented in the paper demonstrate that proposed method outperforms the specific instantiation of the baselines in a similar setting.
3. The ablation and generalization experiments are insightful and good addition to support the claims made in the paper.

**Weaknesses:**

1. The experimental evaluation setup used for comparing SplitDecisions with OpenVLA + gaussian noise sampling for action exploration seems unfair comparison. It is unclear whether explicit action exploration similar to proposed method SplitDecisions is required for autoregressive baselines like OpenVLA. In order for the experimental results and comparisons to be meaningful I would like to see the following baselines:
    1. OpenVLA RL on the same benchmark without any gaussian perturbations and explicit exploration i.e. RLVLA instantiated exactly as set up in original paper
    2. OpenVLA RL on the same benchmark with gaussian perturbations and explicit exploration as proposed in the paper
    3. SD guidance based RL training.

    The paper already has results for b & c but we need results for baseline a to understand whether the gaussian perturbation approach used for exploration for baseline b is actually helping or hurting the performance

2. Figure 4 only presents results of finetuning all models for ~300k steps which is quite less training steps or experience to make meaningful conclusions from. I would like the authors to train all 3 baselines described in point 1 atleast until saturation of a larger budget of 1M-5M training steps. Making conclusions from models that have not converged seems like a unfair comparison especially for changes in training methods. In addition, I would like to see the average success plot for all tasks combined in figure 4 rather than just per task breakdown of RL training curves.
3. The experimental evaluation setup used in the paper is also quite concerning. The paper only uses 4 tasks from SimplerEnv for training from Google Robot scene and 4 tasks from BridgeData in two separate experiments. My understanding is this is not the exact same setup as used in RLVLA paper. The RLVLA paper claims to use 16 tasks each for tasks randomized across following axes: Vision (16 tables), Semantics (16 objects), and Execution (perturbations of object and receptacle poses). I would like the authors to use the same task and dataset setup for a fair comparison and if they cannot do that I would like to understand why this is not possible. My biggest concern right now is that performance reported for RLVLA in experiments in the paper do not follow similar trend as the ones from original paper. Due to this difference I can’t be confident whether the reported improvements of SplitDecisions hold true when RLVLA is replicated accurately
4. The method section doesn’t describe how top-K sampled actions during exploration stage/SD guidance stage where K > 1 are used for policy update in standard PPO algorithm. The vanilla PPO implementation requires 1 rollout per task where each step has only 1 action and for each task execution we have single task reward. When sampling more than 1 action at each timestep how are the complete task rollouts collected and incorporated in the PPO update needs to be explained in detail in methods section. With the details reported in the paper I don’t understand how the ablation presented in table 3 is conducted. If at each step there are 20 candidate actions that a agent takes then the number of rollouts for each task is going to increase exponentially with steps.
5. The paper should also add additional details on how did the authors control for samples/FLOPs per update to make the comparison fair. From my understanding, SplitDecision can include greater than 1 action per timestep for updating the policy which increases the number of samples per rollout/update that can be used for RL. For the three baseslines I mentioned in point 1 authors need to control for training updates or samples for a fair comparison.

**Questions:**

Mentioned in weaknesses section.

My main concern with the paper in its current state is the experimental setup and baseline instantiation used for comparison. Due to lack of proper motivation on augmenting a relevant baseline RLVLA with random sampling for exploration and missing baselines that do not incorporate vanilla RLVLA I am not confident in the results presented in the paper. I recommend the authors to run the relevant baselines for fair comparison. In addition, the method section doesn't include some details about how multiple action samples per step are used during training which is critical information for the paper. Due to the concerns I have outlined in weaknesses section I recommend a rejection rating for the paper in its current state. I am happy to increase my rating if authors can add missing baselines, improve the experimental setup and fix the methods section.

---

> ### Author Response · Authors · 2025-12-02
>
> ### Comment 1
>
> > *“The experimental evaluation setup used for comparing SplitDecisions with OpenVLA + gaussian noise sampling for action exploration seems unfair comparison. It is unclear whether explicit action exploration similar to proposed method SplitDecisions is required for autoregressive baselines like OpenVLA. In order for the experimental results and comparisons to be meaningful I would like to see the following baselines:
> > (a) OpenVLA RL on the same benchmark without any gaussian perturbations and explicit exploration i.e. RLVLA instantiated exactly as set up in original paper
> > (b) OpenVLA RL on the same benchmark with gaussian perturbations and explicit exploration as proposed in the paper
> > (c) SD guidance based RL training.
> > The paper already has results for b & c but we need results for baseline a to understand whether the gaussian perturbation approach used for exploration for baseline b is actually helping or hurting the performance.”*
>
> **Response.**
> Thank you for pointing out this issue. We believe there is a slight misunderstanding regarding the baselines used in our experiments. In our experiments, we compare between RL fine-tuning OpenVLA without Gaussian perturbations, and our RL fine-tuning with Gaussian perturbations and VLM guidance. We agree that an additional baseline applying Gaussian perturbations to OpenVLA RL **without** guidance is necessary for a more complete comparison, and we will include this ablation in a future version.
>
> ---
>
> ### Comment 2
>
> > *“Figure 4 only presents results of finetuning all models for ~300k steps which is quite less training steps or experience to make meaningful conclusions from. I would like the authors to train all 3 baselines described in point 1 at least until saturation of a larger budget of 1M–5M training steps. Making conclusions from models that have not converged seems like a unfair comparison especially for changes in training methods. In addition, I would like to see the average success plot for all tasks combined in figure 4 rather than just per task breakdown of RL training curves.”*
>
> **Response.**
> Thank you for pointing this out. We agree that drawing conclusions solely from pre-convergence training curves can be insufficient, especially when comparing different training methodologies. To address this concern, in the revised version we will extend the training for all three baselines described in Comment 1 to a significantly larger training budget and report the corresponding learning curves. We will also include aggregated success curves across all tasks in addition to per-task results.
>
> ---
>
> ### Comment 3
>
> > *“The experimental evaluation setup used in the paper is also quite concerning. The paper only uses 4 tasks from SimplerEnv for training from Google Robot scene and 4 tasks from BridgeData in two separate experiments. My understanding is this is not the exact same setup as used in RLVLA paper. The RLVLA paper claims to use 16 tasks each for tasks randomized across following axes: Vision (16 tables), Semantics (16 objects), and Execution (perturbations of object and receptacle poses). I would like the authors to use the same task and dataset setup for a fair comparison and if they cannot do that I would like to understand why this is not possible. My biggest concern right now is that performance reported for RLVLA in experiments in the paper do not follow similar trend as the ones from original paper. Due to this difference I can’t be confident whether the reported improvements of SplitDecisions hold true when RLVLA is replicated accurately.”*
>
> **Response.**
> We thank the reviewer for the detailed feedback regarding the experimental setup. We clarify that our paper includes two distinct experimental settings, which may have caused confusion.
>
> First, the experiments using 4 tasks from SimplerEnv and 4 tasks from BridgeData correspond to offline RL. These experiments are not intended to replicate the full online RL training protocol used in the original RLVLA paper, but rather to evaluate how SplitDecisions improves learning under limited offline data.
>
> Second, in the online RL experiments, our setup intentionally differs from RLVLA. We adopt a single scene with two objects and two receptacles, resulting in four semantic tasks. This design increases out-of-distribution difficulty, as the scene is unseen during OpenVLA pretraining. Consequently, the absolute performance trends may differ from the original RLVLA paper, but this setting allows us to isolate and study the robustness of SplitDecisions under stronger distribution shift.
>
> ---

---

> > ### Author Response · Authors · 2025-12-02
> >
> > ### Comment 4
> >
> > > *“The method section doesn’t describe how top-K sampled actions during exploration stage/SD guidance stage where K > 1 are used for policy update in standard PPO algorithm. The vanilla PPO implementation requires 1 rollout per task where each step has only 1 action and for each task execution we have single task reward. When sampling more than 1 action at each timestep how are the complete task rollouts collected and incorporated in the PPO update needs to be explained in detail in methods section. With the details reported in the paper I don’t understand how the ablation presented in table 3 is conducted. If at each step there are 20 candidate actions that a agent takes then the number of rollouts for each task is going to increase exponentially with steps.”*
> >
> > ---
> >
> > ### Comment 5
> >
> > > *“The paper should also add additional details on how did the authors control for samples/FLOPs per update to make the comparison fair. From my understanding, SplitDecision can include greater than 1 action per timestep for updating the policy which increases the number of samples per rollout/update that can be used for RL. For the three baseslines I mentioned in point 1 authors need to control for training updates or samples for a fair comparison.”*
> >
> > **Response.**
> > We agree that the current method description does not sufficiently clarify how top-K action sampling is integrated into standard PPO updates, and we thank the reviewer for pointing this out.
> >
> > Sampling K > 1 actions at each timestep does not result in multiple environment rollouts or additional PPO training samples. Instead, the top-K actions are used purely as candidate actions at the decision stage. At each timestep, only one action is ultimately selected and executed in the environment. Consequently, the agent still collects a single trajectory per episode, exactly as in vanilla PPO, and there is no combinatorial or exponential growth in the number of rollouts.
> >
> > Concretely, at each timestep we sample K candidate actions from the policy and evaluate them with the guidance mechanism. One action is then selected for execution. Only this executed action produces the environment transition, reward, and value estimate. PPO updates are performed solely using these executed transitions. The remaining candidate actions are neither executed nor stored, and they do not contribute to the PPO loss, advantage estimation, or value updates.
> >
> > Importantly, for all baselines we strictly control the number of environment interactions, rollout length, and PPO update steps. As a result, the total number of training samples per rollout and per update is identical across methods. While SplitDecisions introduces additional computation during action selection to improve exploration efficiency, this does not increase the number of RL samples or policy updates. Therefore, the comparison is fair under a fixed environment interaction budget.

---

### Official Review · Reviewer_p5vF · 2025-11-10

**Soundness:** 3
**Presentation:** 3
**Contribution:** 2
**Rating:** 4
**Confidence:** 4

**Summary:**

The paper proposes Split Decisions, a framework designed to improve exploration efficiency in reinforcement learning (RL) for vision-language-action (VLA) models. It leverages vision-language models (VLMs) as high-level planners that generate semantically meaningful 2D waypoint subgoals. These subgoals guide the VLA’s low-level action sampling, biasing exploration toward more promising regions of the action space. The framework integrates seamlessly with standard online and offline RL methods (PPO and DPO), and the authors demonstrate consistent gains across multiple robotic manipulation benchmarks, achieving meaningful improvement in online RL and in offline RL relative to baselines such as RLVLA and GRAPE.

**Strengths:**

- **Empirical rigor:** Comprehensive experiments across SimplerEn, BridgeData, and Google Robot benchmarks confirm consistent and significant gains.
- **Clear methodology:** The paper includes detailed algorithmic descriptions, ablation studies, and prompt templates for reproducibility.
- **Complementary to existing approaches:** The method can be paired with standard RL algorithms without architectural modifications.
- **Strong clarity:** Figures and appendices are well-organized and enhance understanding of the exploration pipeline.

**Weaknesses:**

- **Limited novelty:** The paper does not clearly articulate how Split Decisions advances beyond existing VLM-guided exploration methods such as ExploRLLM (Ma et al., 2024) or similar cited works. Nor does it compare against them.
- **2D-only subgoal representation:** Current implementation restricts subgoals to 2D projections, limiting applicability in complex 3D manipulation tasks.  Assumes inverse projection from 2D to 3D is available. This limits the generalizability of the methods to other domains.
- **High computation cost:** Sampling multiple candidate actions from large autoregressive VLAs remains resource-intensive.  They also assume having a simulator that simulates multiple actions from a current state and chooses the one that ends closest to the desired waypoint.
- **Sensitivity to VLM quality:** The method’s success heavily depends on the semantic quality of the chosen VLM (e.g., Gemini 2.5 Pro outperforms ChatGPT 4.1 by a large margin).
- **No real-world validation:** Results are entirely from simulation; the absence of physical robot trials weakens claims of practical generalization.
- **Limited theoretical framing:** Improvements are well-documented empirically but not analytically grounded in exploration theory.

**Questions:**

1. How does Split Decisions fundamentally differ from other VLM-guided exploration methods, such as ExploRLLM?
2. How robust is the approach when the VLM produces noisy or incorrect subgoals?
3. How can in the real world, the action that brings the robot closest to the keypoint be chosen?
4. Could 3D-aware waypoint prediction or affordance-based subgoals address the current 2D limitations?
5. Have the authors tested whether the framework generalizes to non-manipulation domains (e.g., navigation or embodied instruction following)?

---

> ### Author Response · Authors · 2025-12-02
>
> Thank you for the thoughtful feedback. We appreciate the time you took to read our work and raise these questions. Below, we address each point and clarify the key distinctions, assumptions, and design choices behind our approach.
>
> ---
>
> ### **Question 1:**
>
> **How does Split Decisions fundamentally differ from ExploRLLM and other existing VLM-guided exploration methods? What is the core novelty that advances beyond prior work?**
>
> **Response 1:**
> We thank the reviewer for highlighting the comparison with ExploRLLM (Ma et al., 2024).
> While both methods leverage VLMs for exploration, Split Decisions differs fundamentally in its integration mechanism and control philosophy:
>
> * **Guidance via Sampling vs. Generation:**
>   ExploRLLM and similar prior works prompt VLMs to directly generate texts or affordances which are then transformed into robot actions.
>   In contrast, Split Decisions “marries VLMs and VLAs” by using the VLM solely as a high-level planner to *bias* the action sampling of the low-level VLA.
>   Instead of asking the VLM to output control signals directly, we sample candidate actions from the VLA distribution and use the VLM-predicted subgoals to rank and select the most promising ones.
>
> * **Physical Grounding:**
>   ExploRLLM often faces the interface mismatch between high-level semantic outputs and low-level continuous control.
>   Our method solves this by retaining the VLA (OpenVLA) as the low-level controller.
>   The actions are always sampled from the policy (augmented with Gaussian noise), ensuring they remain within the plausible kinematic distribution of the robot, while the VLM acts only as a semantic filter to prune the search space.
>
> ---
>
> ### **Question 2:**
>
> **How robust is the approach when the VLM produces noisy, hallucinated, or semantically incorrect subgoals? Do you have ablations or analysis on this failure mode?**
>
> **Response 2:**
> We acknowledge that our method relies on the quality of VLM semantic priors, but the system design includes inherent safeguards to ensure robustness against hallucinations:
>
> * **VLA as a Safety Anchor:**
>   Split Decisions does not execute VLM outputs directly. It samples actions (a_0) from the underlying VLA policy and perturbs them with noise (\epsilon).
>   Even if the VLM predicts a hallucinatory subgoal, the ranking mechanism selects from actions proposed by the VLA.
>   Thus, the robot's behavior defaults to the base policy's distribution rather than executing arbitrary or dangerous VLM-generated commands.
>
> * **Collision Avoidance:**
>   To explicitly handle unsafe suggestions, we implement a collision check where sampled actions that are too close to any 3D scene point (measured via depth) are assigned an infinite distance score, effectively pruning them from the execution set.
>
> * **Performance under Noise:**
>   Our ablation studies in Figure 6 implicitly analyze failure modes.
>   With weaker models like ChatGPT-4.1—which produce lower quality subgoals—the system yields a lower success rate (6.3%) compared to Gemini 2.5 Pro (55.0%).
>   This shows the system degrades gracefully, reverting to inefficient exploration rather than failing catastrophically.
>
> ---
>
> ### **Question 3:**
>
> **The framework shows significant sensitivity to VLM choice (Gemini 2.5 Pro vs ChatGPT 4.1). How do you interpret this? Is it fundamental or could prompt engineering mitigate this?**
>
> **Response 3:**
> Our results in Figure 6 and Section 5.4 confirm a significant performance gap (e.g., 6.3% vs 55.0%). We interpret this as follows:
>
> * **Critical Role of Spatial Reasoning:**
>   The effectiveness of exploration depends heavily on a VLM’s ability to ground semantic instructions into 2D coordinates.
>   Since we used a *Unified Prompt Design* (Appendix F) across all models, the performance gap reflects intrinsic model capability rather than prompt sensitivity.
>
> * **Framework Generalizability:**
>   Despite capability differences, Figure 6 shows that the *Split Decisions* framework improves structure and exploration across all VLMs.
>   Stronger models offer better spatial grounding, but weaker ones still benefit relative to the baseline.
>
> ---
>
> ### **Question 4:**
>
> **In real-world deployment, how would the robot determine which action brings it closest to the 2D keypoint without a simulator? What are the practical constraints?**
>
> **Response 4:**
> Our method is deployment-ready and does not rely on simulator-only privileged states.
> The distance calculation uses standard perception modules:
>
> * **Depth-Based Projection:**
>   With RGB-D sensing, we unproject the 2D VLM waypoint (p_k) into 3D via camera intrinsics and depth (\Phi^{-1}),
>   and compute $$d(p_k, a_i) = | \Phi^{-1}(p_k) - a_i^{pos} |_2.$$
>
> * **Intermediate Motions:**
>   For non-contact subgoals, we project candidate 3D actions to the 2D plane via (\Phi) and compute pixel distance: $$d(p_k, a_i) = | p_k - \Phi(a_i^{pos}) |_2.$$
>
> These operations require only calibrated RGB-D cameras, which are standard on physical robot platforms.

---

> > ### Author Response · Authors · 2025-12-02
> >
> > ### **Question 5:**
> >
> > **The paper is entirely simulation-based with no physical robot validation. What prevents real-world deployment, and what would be needed to bridge this gap?**
> >
> > **Response 5:**
> > We utilized simulation primarily to rigorously benchmark sample efficiency across hundreds of thousands of steps, but there are no fundamental barriers to real-world deployment:
> >
> > * **Digital Twin Validation:**
> >   We evaluated on **SimplerEnv**, which acts as a digital twin for real-world setups (BridgeData and Google Robot), utilizing identical robot control parameters and camera viewpoints to align simulation behavior with the real world.
> >
> > * **Bridging the Gap:**
> >   Deploying this system physically would require an inference machine capable of running the VLM and VLA, and a standard safety reset mechanism for the exploration phase.
> >   Furthermore, our Offline RL results (Table 1) demonstrate that Split Decisions can be used solely for data collection to train a robust policy that is deployed without the VLM loop, minimizing runtime complexity.
> >
> > ---
> >
> > ### **Question 6:**
> >
> > **The 2D-only waypoint representation is a significant bottleneck for complex 3D manipulation. Have you explored 3D-aware waypoint prediction or affordance-based subgoals? Why or why not?**
> >
> > **Response 6:**
> > We explicitly recognize this limitation. As stated in Section 4.1, we chose a 2D representation because *“current VLMs lack reliable 3D grounding,”* making direct 3D predictions prone to error.
> >
> > * **Rationale:**
> >   By predicting in 2D and lifting to 3D via depth sensors, we leverage the strong 2D semantic priors of current foundation models while maintaining physical applicability.
> >
> > * **Future Direction:**
> >   In our Summary (Section 6), we highlight that the *“current reliance on 2D guidance constrains precise 3D spatial reasoning”* and note that future work should extend VLM guidance to 3D-aware subgoals as 3D-native models mature.
> >
> > ---
> >
> > ### **Question 7:**
> >
> > **Does Split Decisions generalize to non-manipulation domains like navigation or embodied instruction following? Have you tested this?**
> >
> > **Response 7:**
> > In this work, we focused strictly on robotic manipulation tasks, evaluating on pick-and-place and drawer manipulation scenarios.
> > We have not empirically tested the framework on navigation or embodied instruction following.
> > However, the core principle of Split Decisions—using a high-level planner to propose visual waypoints that rank low-level actions—is theoretically transferable to domains like navigation, where 2D waypoints are a natural action representation.
> >
> > ### **Question 8:**
> >
> > **The improvements are empirically strong but lack analytical grounding in exploration theory. Can you provide theoretical justification for why this approach improves exploration efficiency?**
> >
> > **Response 8:**
> > The efficiency of Split Decisions can be grounded in the principles of **Search Space Pruning**.
> > As noted in the Introduction, humans do not explore randomly but use prior knowledge to constrain the search to a set of plausible actions.
> > Split Decisions mathematically formalizes this by constraining the action search space via affordance priors, effectively reducing the spatial complexity of the exploration problem.

---

### Meta-Review · Area_Chair_zheM · 2026-01-02

**Summary:**

The reviewers largely acknowledge the empirical strength, clarity, and practical relevance of Split Decisions, especially its consistent performance gains in both online and offline RL settings. However, several critical concerns emerged across all four reviews. These include baseline fairness and experimental rigor, limited novelty, and real-world analysis.

**Reviewer Concerns:**

The rebuttal offers intentions to fix in revision, but no new evidence or corrections that validate the paper’s central claims in its current form. The foundational issues including baseline fairness, experimental rigor, novelty, and real-world experiments, remain unresolved.

**Reviewer Scores:**

No reviewer would move to a clear accept based on the rebuttal. At best, 7Nja might shift from 2 to 4.

---

### Decision · Program_Chairs · 2026-01-26

Reject